# Discovery of ZrCoBi based half Heuslers with high thermoelectric conversion efficiency

Hangtian Zhu[1], Ran He[1,2], Jun Mao [1,3], Qing Zhu[1], Chunhua Li[4], Jifeng Sun[5], Wuyang Ren[1,6], Yumei Wang[7], Zihang Liu[1], Zhongjia Tang[8], Andrei Sotnikov[2], Zhiming Wang[6], David Broido[4], David J. Singh[5], Gang Chen [9], Kornelius Nielsch[2] & Zhifeng Ren[1]

Thermoelectric materials are capable of converting waste heat into electricity. The dimensionless figure-of-merit (ZT), as the critical measure for the material's thermoelectric performance, plays a decisive role in the energy conversion efficiency. Half-Heusler materials, as one of the most promising candidates for thermoelectric power generation, have relatively low ZTs compared to other material systems. Here we report the discovery of *p*-type ZrCoBi-based half-Heuslers with a record-high ZT of ~1.42 at 973 K and a high thermoelectric conversion efficiency of ~9% at the temperature difference of ~500 K. Such an outstanding thermoelectric performance originates from its unique band structure offering a high band degeneracy ($N_v$) of 10 in conjunction with a low thermal conductivity benefiting from the low mean sound velocity ($v_m$ ~2800 m s$^{-1}$). Our work demonstrates that ZrCoBi-based half-Heuslers are promising candidates for high-temperature thermoelectric power generation.

[1] Department of Physics and Texas Center for Superconductivity, University of Houston, Houston, TX 77204, USA. [2] Institute for Metallic Materials, IFW-Dresden, Dresden 01069, Germany. [3] Department of Mechanical Engineering, University of Houston, Houston, TX 77204, USA. [4] Department of Physics, Boston College, Chestnut Hill, MA 02467, USA. [5] Department of Physics and Astronomy, University of Missouri, Columbia, MO 65211, USA. [6] Institute of Fundamental and Frontier Sciences, University of Electronic Science and Technology of China, 610054 Chengdu, China. [7] Beijing National Laboratory for Condensed Matter Physics, Institute of Physics, Chinese Academy of Sciences, P.O. Box 603, 100190 Beijing, China. [8] Department of Chemistry, University of Houston, Houston, TX 77204, USA. [9] Department of Mechanical Engineering, Massachusetts Institute of Technology, Cambridge, MA 02139, USA. These authors contributed equally: Hangtian Zhu, Ran He, Jun Mao. Correspondence and requests for materials should be addressed to Z.R. (email: zren@uh.edu)

Thermoelectric generators enable a direct energy conversion from heat to electricity[1,2]. This solid-state energy conversion technique has advantages of reliability, simplicity, compactness, and environmentally friendliness. However, the application of thermoelectric modules is currently limited to niche market due to the relatively low efficiency comparing to the traditional heat engines. The conversion efficiency of the thermoelectric modules is jointly determined by the Carnot efficiency, as well as the material's figure-of-merit (ZT):

$$\eta = \left( \frac{T_{hot} - T_{cold}}{T_{hot}} \right) \left[ \frac{\sqrt{1 + ZT_m} - 1}{\sqrt{1 + ZT_m} + \left( \frac{T_{cold}}{T_{hot}} \right)} \right] \quad (1)$$

where $T_{hot}$ is the hot-side temperature, $T_{cold}$ is the cold-side temperature, and $T_m$ is the average temperature. ZT is the thermoelectric figure-of-merit, which is the critical measure for the materials' performance that defined as:

$$ZT = \frac{S^2 \sigma}{\kappa_L + \kappa_e} T \quad (2)$$

where $S$, $\sigma$, $\kappa_L$, $\kappa_e$, and $T$ are the Seebeck coefficient, electrical conductivity, lattice thermal conductivity, electronic thermal conductivity, and absolute temperature, respectively[3–6]. At a given temperature difference ($T_{hot}$–$T_{cold}$), the improvements in thermoelectric conversion efficiency entirely rely upon the ZT enhancement. Therefore, improving the performance of existing materials and identifying new compounds with intrinsically high ZT are two basic concepts in the research of thermoelectric materials.

However, simultaneous optimization of the thermoelectric transport parameters remains to be a grand challenge owing to their intricate inter-dependences[7]. The $S$, $\sigma$, and $\kappa_e$ are closely correlated with each other via the carrier concentration, while $\kappa_L$ is relatively independent. Therefore, improving the power factor ($S^2\sigma$) and reducing the lattice thermal conductivity are two main strategies for enhancing the thermoelectric performance. On the one hand, the power factors can be optimized by tuning the carrier concentration, and further enhancement can be achieved by band engineering[8–10], modulation doping[11], introducing the resonant level[12], and tuning the carrier scattering mechanism[13–15]. Among these approaches, band engineering via increasing the degenerate band valleys ($N_v$), either by alloying or exploiting the temperature dependence of the electronic bands, has been demonstrated to be particularly effective in enhancing the power factor. In this case, a high electrical conductivity can be obtained with the presence of multiple conducting channels enabled by the high number of band valleys. In the meanwhile, the Seebeck coefficient can still be maintained since the high electrical conductivity do not involve any increase in the carrier concentration.

On the other hand, reduction of the lattice thermal conductivity has been proven quite effective in enhancing the ZT. According to the kinetic theory, $\kappa_L = \frac{1}{3} C_v \nu_{ph} l$, where $C_v$ is the heat capacity, $\nu_{ph}$ is the phonon velocity, and $l$ is the phonon mean free path[16]. Extensive results have demonstrated that shortening the phonon mean-free-path via phonon scattering by microstructural defects[17–20] and nanostructures[21–23] can noticeably reduce the lattice thermal conductivity. In addition, phonon velocity as another important parameter for lattice thermal conductivity can also be tailored for phonon engineering. Usually, the phonon velocity is simply approximated by the low frequency sound velocity ($\nu \propto \sqrt{B/\delta}$), where $B$ is the elastic modulus and $\delta$ is the density of the compound[24]. Therefore, sound velocity, which is closely associated with the crystal structure, chemical composition, and bonding, can play a decisive role in the lattice

thermal conductivity. More specifically, the materials with a low sound velocity usually tend to have a low lattice thermal conductivity[25]. Therefore, a novel compound that simultaneously possesses a high band degeneracy (i.e., high power factor) in combination with an intrinsically low sound velocity (i.e., low thermal conductivity) is very likely to demonstrate a promising thermoelectric performance.

Among the various thermoelectric materials, half-Heusler compounds, with large power factors[26], robust mechanical properties[27], and excellent thermal stabilities[28], have been recently recognized as one of the most promising candidates for high temperature thermoelectric power generation[29–33]. However, due to the relative high lattice thermal conductivity (usually on the magnitude of ~10 W m$^{-1}$ K$^{-1}$ for the pristine compounds), the ZTs (especially the average ZTs) of the state-of-the-art half-Heuslers remains relatively low comparing to other well-established material systems. Consequently, future development of the half-Hesulers thermoelectric module hinges largely on identifying a new compound with high thermoelectric performance.

Here we report the discovery of $p$-type ZrCoBi-based half-Heuslers that possess a high band degeneracy ($N_v = 10$) in conjunction with a lowest mean sound velocity ($\nu_m$ ~2800 m s$^{-1}$) among the state-of-the-art half-Heuslers[25,34]. Benefiting from the combination of the appealing electronic and thermal properties, a record-high peak ZT of ~1.42 at 973 K can be achieved. Such an exceptionally high thermoelectric performance is further validated by the efficiency measurement and a high thermoelectric conversion efficiency of ~9% is achieved at the temperature difference of ~500 K. Our work demonstrates that ZrCoBi-based half-Heuslers are quite promising for high-temperature thermoelectric power generation. Importantly, the Bi-based half-Heuslers, which have long been ignored for thermoelectric application, open up a new avenue for designing advanced half-Heusler thermoelectric materials in the future.

## Results

**High thermoelectric performance of ZrCoBi.** To demonstrate the high thermoelectric performance of the ZrCoBi-based half-Heuslers, comparison of the temperature-dependent ZT between ZrCoBi$_{0.65}$Sb$_{0.15}$Sn$_{0.20}$ and the state-of-the-art $p$-type half-Heuslers (e.g., HfCoSb-baseds, ZrCoSb-based, and NbFeSb-based half-Heuslers)[26,35–37] is shown in Fig. 1a. Clearly, ZrCoBi$_{0.65}$Sb$_{0.15}$Sn$_{0.20}$ outperforms all the other $p$-type half-Heuslers in the whole temperature range and a record-high peak ZT of ~1.42 at 973 K can be achieved. The average ZT is further calculated by the integration method in the temperature of 300 to 973 K, where ZrCoBi$_{0.65}$Sb$_{0.15}$Sn$_{0.20}$ demonstrates the highest average ZT of ~0.81, and it is only ~0.69 for Nb$_{0.88}$Hf$_{0.12}$FeSb, ~0.57 for Nb$_{0.8}$Ti$_{0.2}$FeSb, ~0.54 for Hf$_{0.44}$Zr$_{0.44}$Ti$_{0.12}$Sb$_{0.8}$Sn$_{0.2}$, and ~0.53 for Hf$_{0.8}$Ti$_{0.2}$CoSb$_{0.8}$Sn$_{0.2}$.

**High band degeneracy leads to high power factor.** To understand the origin for such a high thermoelectric performance of ZrCoBi-based half-Heuslers, the first-principles calculation on the band structure of ZrCoBi (Fig. 2a) was employed to evaluate its electronic thermoelectric performance. The valence band maxima (VBM) locates at Γ point (marked by blue color), while the valence bands at L point (marked by red color) show a negligible energy difference (ΔE) of ~0.001 eV lower than that of Γ point. The valence bands converge at Γ point and split slightly at L point due to spin-orbit coupling effect as shown in the Supplementary Fig. 1. Due to the negligible energy difference, all the valence bands that converge at L and Γ points will contribute jointly to the hole-transport. Since the two bands converging at L point give

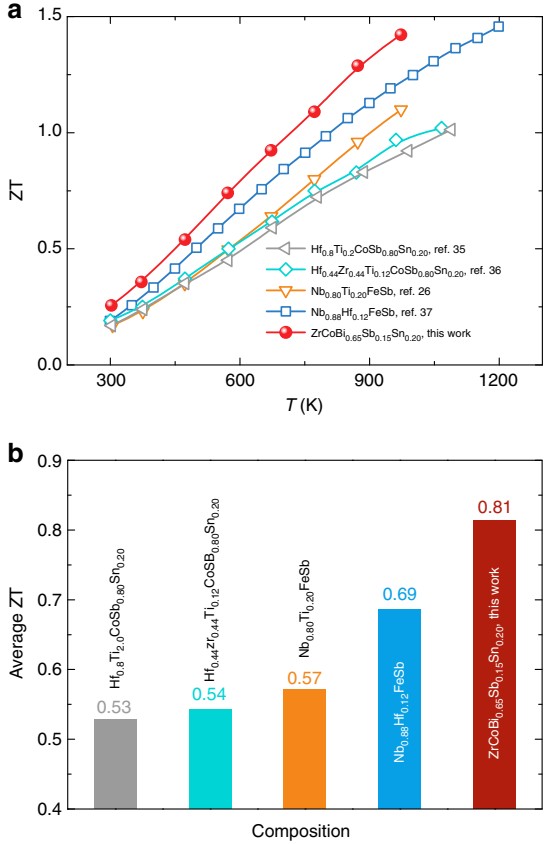

**Fig. 1** Thermoelectric figure-of-merit ZT. Comparison of temperature-dependent ZT values (**a**) and average ZTs (**b**) between *p*-type ZrCoBi$_{0.65}$Sb$_{0.15}$Sn$_{0.2}$ and the reported state-of-the-art *p*-type half-Heuslers

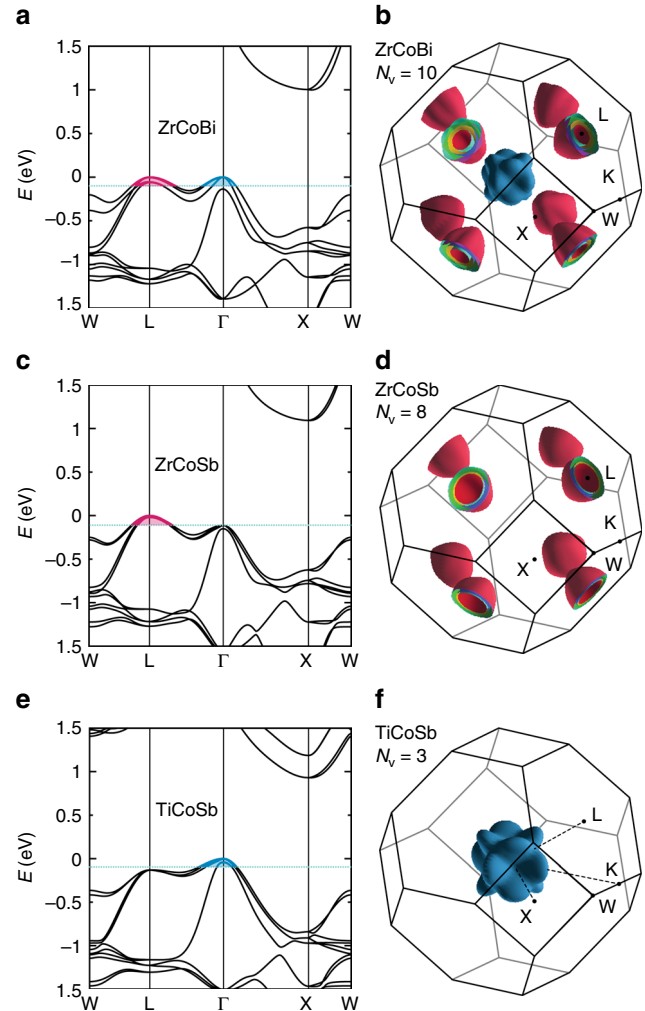

**Fig. 2** First-principle calculation of band structure. Calculated band structures of ZrCoBi (**a**), ZrCoSb (**c**), and TiCoSb (**e**). The blue lines represent energy level of 0.1 eV below VBM. The corresponding iso-energy surfaces at 0.1 eV below VBM in Brillouin zone of ZrCoBi (**b**), ZrCoSb (**d**), and TiCoSb (**f**)

a band degeneracy of 8 and those at Γ point provide an additional band degeneracy of 2, therefore in total yielding a high band degeneracy of 10 for *p*-type ZrCoBi (Fig. 2b). To highlight such a high band degeneracy in ZrCoBi, band structures for the iso-structural half-Heuslers ZrCoSb (Fig. 2c) and TiCoSb (Fig. 2e) were also calculated for comparison. In contrast, there is an appreciable energy difference of L and Γ points for ZrCoSb (ΔE ~0.13 eV) and TiCoSb (ΔE ~0.11 eV), which means only the valence bands at one of the points will contribute to the hole-transport. To better illustrate the differences in band degeneracy among the three compounds, iso-energy surface at 0.1 eV below VBM is plotted (Fig. 2b, d, f). The band degeneracy equals 8 for ZrCoSb and only 3 for TiCoSb, both of which are noticeably lower than that of ZrCoBi. According to the above-mentioned relationship between the band degeneracy and power factor, it strongly suggests that *p*-type ZrCoBi-based compounds could demonstrate a quite promising electronic thermoelectric performance.

In this work, ZrCoBi-based materials are synthesized by the ball-milling and hot-pressing technique (details can be found in the methods section). All the prepared specimens demonstrate a single half-Heusler phase as shown in the Supplementary Fig. 2. Figure 3a shows the electrical conductivity of ZrCoBi$_{1-x}$Sn$_x$ ($x$ = 0, 0.05, 0.10, 0.15, and 0.20), where a monotonic increase of electrical conductivity with the Sn concentration can be observed. The room temperature electrical conductivity is ~1.14 × 10$^3$ Ohm$^{-1}$ m$^{-1}$ for ZrCoBi and it is ~1.66 × 10$^5$ Ohm$^{-1}$ m$^{-1}$ for ZrCoBi$_{0.8}$Sn$_{0.2}$. The enhancement in electrical conductivity should be mainly attributed to the effectively increased Hall carrier concentration ($n_H$), as shown in the Supplementary Fig. 3.

The almost linear increase of Hall carrier concentration with respect to the Sn concentration (it is ~0.75 × 10$^{21}$ cm$^{-3}$ for ZrCoBi$_{0.95}$Sn$_{0.05}$ and ~2.75 × 10$^{21}$ cm$^{-3}$ for ZrCoBi$_{0.8}$Sn$_{0.2}$) demonstrates the high doping efficiency of Sn in *p*-type ZrCoBi. Similarly, the effectiveness of Sn as a *p*-type dopant was also reported in (Hf, Zr, Ti)CoSb[35,36]. Figure 3b shows the temperature-dependent Seebeck coefficient of ZrCoBi$_{1-x}$Sn$_x$. It is noteworthy that the pristine ZrCoBi shows an intrinsic *n*-type transport characteristic and Sn-doping (Sn concentration as low as ~5%) successfully converts it into fully *p*-type. At relatively low Sn concentration ($x$ = 0.05), the bipolar conduction can be observed at high temperature and it disappears when the Sn concentration is increased.

By optimizing the Sn concentration, high power factors can be obtained for ZrCoBi$_{1-x}$Sn$_x$ (Fig. 3c). The room temperature power factor is ~25 μW cm$^{-1}$ K$^{-2}$ and the peak power factor reaches ~40 μW cm$^{-1}$ K$^{-2}$ for ZrCoBi$_{0.80}$Sn$_{0.20}$. In addition, Sn-doped TiCoSb and ZrCoSb (both were prepared in this work with identical approach) with similar Hall carier concentration are also ploted for comparison. As shown in Fig. 3c, ZrCoBi$_{0.75}$Sn$_{0.15}$ and ZrCoBi$_{0.80}$Sn$_{0.20}$ show noticeably higher power factors than that of Sn-doped ZrCoSb and TiCoSb. Band-degeneracy-dependent

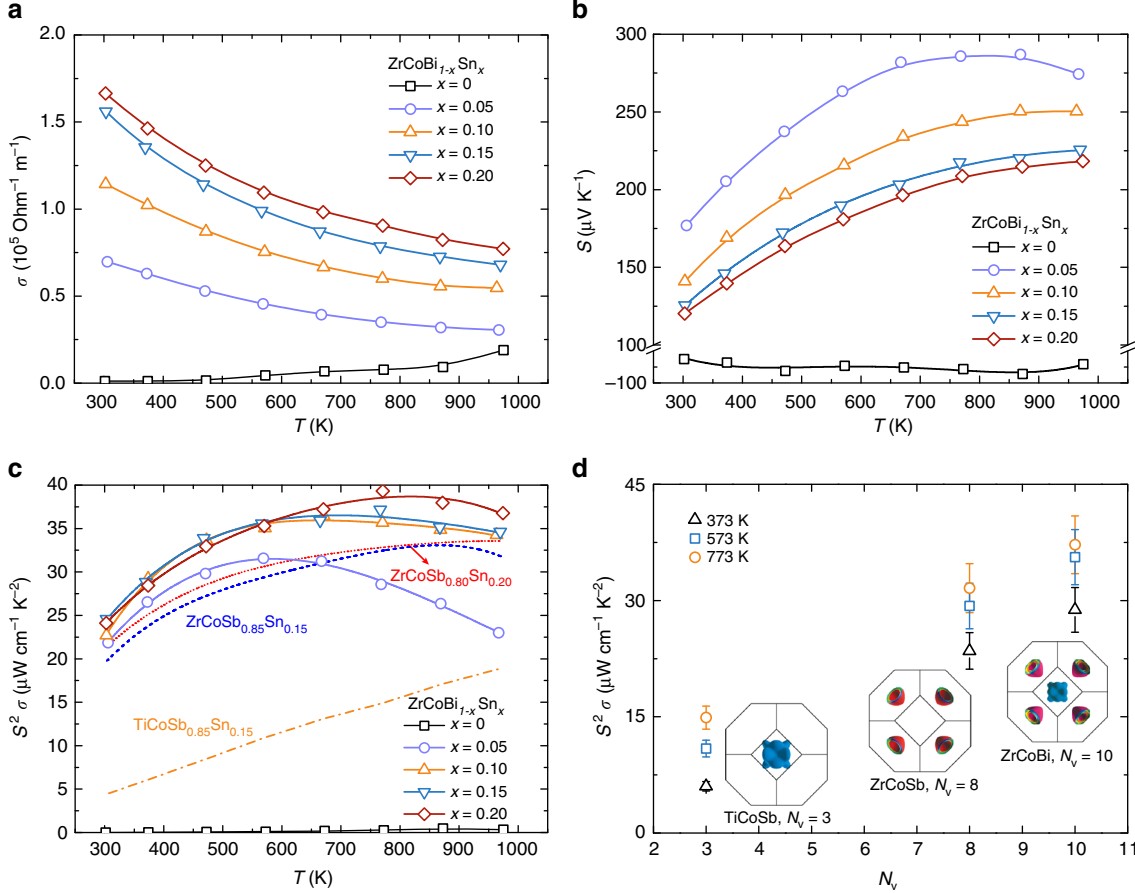

**Fig. 3** Electrical properties of ZrCoBi$_{1-x}$Sn$_x$. Temperature-dependent electrical conductivity (**a**), Seebeck coefficient (**b**), and power factor (**c**) of ZrCoBi$_{1-x}$Sn$_x$ ($x = 0$, 0.05, 0.10, 0.15, and 0.20). **d** Band-degeneracy-dependent power factors for 15% Sn doped TiCoSb, ZrCoSb, and ZrCoBi at different temperatures with carrier concentration of $1.62 \times 10^{21}$, $1.47 \times 10^{21}$, and $2.20 \times 10^{21}$ cm$^{-3}$, respectively

power factor at different temperatures is further plotted for the three compounds, as shown in Fig. 3d. The power factor increases monotonically with the band degeneracy at all of the temperatures. This unambiguously demonstrates that band degeneracy plays a pivotal role in the power factor. In other words, the high power factor achieved in ZrCoBi-based compounds should be mainly ascribed to the high band degeneracy for this compound as indicated by the theoretical calculations (Fig. 2b).

**Low sound velocity leads to low lattice thermal conductivity**. As mentioned above, sound velocity plays a vital role in the lattice thermal conductivity. The relationship between the Young's modulus ($E$) and the mean sound velocity ($v_m$) for the state-of-the-art half-Heuslers[25,27,34,38] is shown in Fig. 4a. Compared to the Sn-based and Sb-based half-Heuslers (NbFeSb, $M$NiSn, and $M$CoSb, where $M =$ Hf, Zr, Ti), the ZrCoBi-based compounds possess the lowest mean sound velocity $\sim 2850$ m s$^{-1}$ (details can be found in the Supplementary Table 1) and Young's modulus. Such a low mean sound velocity and Young's modulus originate from the weaker chemical bonding and heavy atomic mass of Bi. For the ZrCoBi-based compounds, the strong relativistic effect of Bi contracts the 6 s shell and increases its inertness for bonding. Therefore, the low mean sound velocity and Young's modulus will jointly contribute to an intrinsically low lattice thermal conductivity for ZrCoBi.

The temperature-dependent thermal conductivities of ZrCoBi$_{1-x}$Sn$_x$ are shown in Fig. 4b. In addition, the thermal conductivities of the undoped TiCoSb[39], ZrCoSb[40], and NbFeSb[26] are also

plotted, where the pristine ZrCoBi shows a much lower thermal conductivity compared to the other half-Heuslers. The room temperature thermal conductivity is $\sim 19$ W m$^{-1}$ K$^{-1}$ for TiCoSb, $\sim 19$ W m$^{-1}$ K$^{-1}$ for ZrCoSb, and $\sim 17$ W m$^{-1}$ K$^{-1}$ for NbFeSb, but $\sim 9$ W m$^{-1}$ K$^{-1}$ for ZrCoBi, which is only half of the other $p$-type half-Heuslers. Importantly, the thermal conductivity of ZrCoBi$_{1-x}$Sn$_x$ decreases noticeably with Sn concentration, which should be mainly ascribed to the reduction in the lattice thermal conductivity. As shown in Fig. 4c, the lattice thermal conductivity of ZrCoBi$_{1-x}$Sn$_x$ is greatly suppressed with the increase of Sn concentration. The room temperature lattice thermal conductivity of ZrCoBi is $\sim 9$ W m$^{-1}$ K$^{-1}$ but only $\sim 2.6$ W m$^{-1}$ K$^{-1}$ for ZrCoBi$_{0.80}$Sn$_{0.20}$, where a reduction of $\sim 71\%$ is achieved after Sn doping. It is noteworthy that the minimum lattice thermal conductivity of ZrCoBi$_{0.80}$Sn$_{0.20}$ can reach as low as $\sim 1.6$ W m$^{-1}$ K$^{-1}$ at 973 K. Such a significant phonon scattering by Sn-doping should be mainly attributed to the substantial atomic mass difference between Sn (atomic weight: $\sim 118.71$) and Bi atoms (atomic weight: $\sim 208.98$) that leads to an intense point defect scattering. The accumulated lattice thermal conductivities with respect to the phonon mean-free-path for ZrCoBi and ZrCoBi$_{0.80}$Sn$_{0.20}$ are calculated and shown in Fig. 4d. The calculated lattice thermal conductivity of ZrCoBi$_{0.80}$Sn$_{0.20}$ is indeed much lower compared to that of ZrCoBi. The individual contributions from acoustic and optical phonons are marked by different colors. Clearly, the significantly reduced acoustic phonon contribution leads to the much lower thermal conductivity of ZrCoBi$_{0.80}$Sn$_{0.20}$. Such a significant scattering of acoustic phonon should be mainly attributed to the point defect scattering induced by Sn-doping.

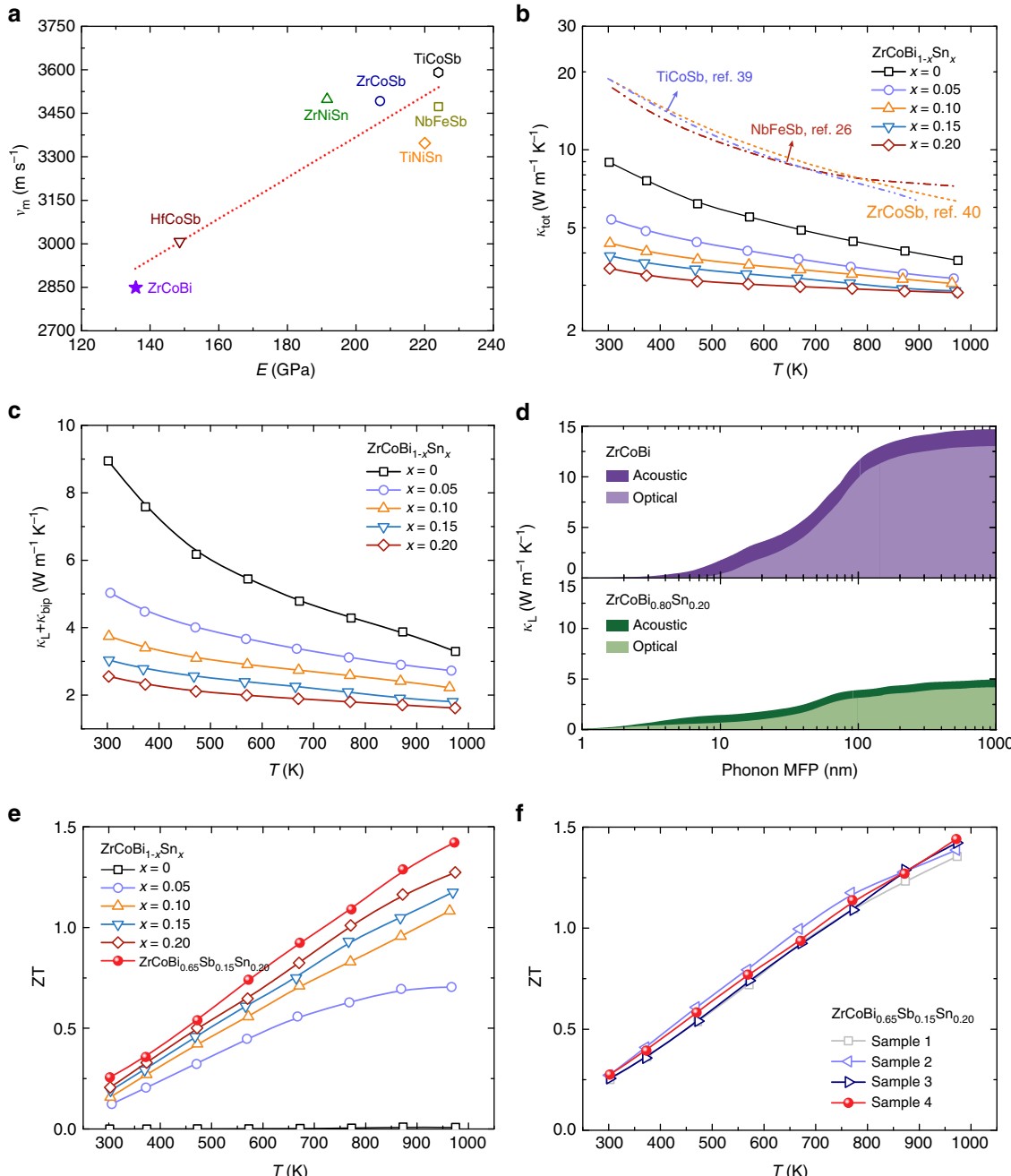

**Fig. 4** Thermal conductivities and ZTs of ZrCoBi$_{1-x}$Sn$_x$. **a** Relationship between Young's modulus ($E$) and mean sound velocity ($v_m$). Temperature-dependent thermal conductivity (**b**) and lattice thermal conductivity (**c**) of ZrCoBi$_{1-x}$Sn$_x$. **d** Accumulated room temperature lattice thermal conductivity of ZrCoBi and ZrCoBi$_{0.8}$Sn$_{0.2}$ with respect to phonon mean-free-path. The notations of acoustic and optical represent acoustic and optical branches of phonons, respectively. **e** Temperature-dependent ZT of ZrCoBi$_{1-x}$Sn$_x$ and ZrCoBi$_{0.65}$Sb$_{0.15}$Sn$_{0.20}$. **f** Reproducibility of the thermoelectric performance of ZrCoBi$_{0.65}$Sb$_{0.15}$Sn$_{0.20}$

Meanwhile, a substantially reduced grain size after Sn doping has also been observed and the average grain size is ~250 nm for ZrCoBi$_{0.80}$Sn$_{0.20}$ (Supplementary Fig. 4). The reduction in grain size should be attributed to the suppressed grain growth during the hot-pressing process. Such a reduced average grain size is beneficial for inducing additional grain boundary scattering to decrease the latttice thermal conductivity. However, as shown in Fig. 4d, the phonon with mean-free-path lower than 100 nm dominates the lattice thermal conductivity. Therefore, compared to the alloying scattering, the grain boundary scattering plays a secondary role in reducing the thermal conductivity of ZrCoBi-based materials.

Owing to the simultaneously enhanced power factor and reduced thermal conductivity via Sn doping, ZT can be noticeably improved in ZrCoBi$_{1-x}$Sn$_x$ (as shown in Fig. 4e). A high peak ZT of ~1.3 at 973 K can be achieved by ZrCoBi$_{0.80}$Sn$_{0.20}$. However, according to the composition-dependent lattice constant of ZrCoBi$_{1-x}$Sn$_x$ (Supplementary Fig. 5), the maximum solubility of Sn at Bi site of ZrCoBi is ~20%. Therefore, in order to further minimize the lattice thermal conductivity, Sb alloying at the Bi site is conducted based upon the best composition of ZrCo-Bi$_{0.80}$Sn$_{0.20}$. As shown in the Supplementary Fig. 6d, e, a noticeable reduction in thermal conductivity can be successfully achieved by Sb alloying. The room temperature lattice thermal

conductivity is ~2.6 W m$^{-1}$ K$^{-1}$ for ZrCoBi$_{0.80}$Sn$_{0.20}$ and it is ~2.2 W m$^{-1}$ K$^{-1}$ for ZrCoBi$_{0.65}$Sb$_{0.15}$Sn$_{0.20}$, an additional reduction of 15%. In the meanwhile, the power factors of ZrCoBi$_{0.80-y}$Sb$_y$Sn$_{0.20}$ remain quite similar (Supplementary Fig. 6c). Collectively, an even higher peak ZT of ~1.42 at 973 K is achieved in ZrCoBi$_{0.65}$Sb$_{0.15}$Sn$_{0.20}$. To prove the reproducibility of the high performance of ZrCoBi$_{0.65}$Sb$_{0.15}$Sn$_{0.20}$, four samples were prepared from different batches and quite comparable results were obtained, as shown in Fig. 4f. Detailed thermoelectric properties of the four samples are shown in the Supplementary Fig. 7.

**High thermoelectric conversion efficiency**. To further validate the high thermoelectric performance of ZrCoBi$_{0.65}$Sb$_{0.15}$Sn$_{0.20}$, heat-to-electricity conversion efficiency ($\eta$) and output power density ($\omega$) were measured on a single-leg device with a home-made system (Fig. 5a and Supplementary Fig. 8). The thermoelectric material (will be referred as leg in the following) was polished to the size of $1.5 \times 2.4$ mm$^2$ in cross-section and ~ 4.65 mm in thickness. The cold side of the leg was electroplated with copper, nickel, and gold layers consequently, then soldered (In$_{52}$Sn$_{48}$, melting point 391 K) to copper plate, and the hot side of the leg was directly brazed (Ag$_{56}$Cu$_{22}$Zn$_{17}$Cd$_5$, liquidus point 923 K) with copper plate. The temperature of cold side was maintained by water circulation and temperature of hot side was controlled by PID. The experiments were conducted under high vacuum (below 10$^{-6}$ mbar) to reduce the heat conduction. To measure conversion efficiency ($\eta$), the input power from hot side

($Q_{in}$) and the generated power ($P$) were measured at the same time. The direct measurement of $Q_{in}$ is of great challenge due to the heavy heat loss at high temperature. According to Fourier's Law, a bulk polycrystalline graphite with measured geometry and thermal conductivity was placed below cold side to measure the heat flow out of cold side ($Q_{out}$). The thermal conductivity of the bulk polycrystalline graphite was confirmed before in the same way as described in the methods section. In order to measure temperature differences of the leg and graphite bulk, K-type thermocouples were embedded at the interfaces. It should be noticed that the hot-side temperature of graphite can be regarded as the cold-side temperature of the leg if the setup is working under a large pressure as shown in Supplementary Fig. 8. The total $Q_{in}$ includes $Q_{out}$, $P$ and radiation loss from the leg ($Q_{rad}$). Therefore, the conversion efficiency ($\eta$) can be written as the following:

$$\eta = \frac{P}{Q_{in}} = \frac{P}{Q_{out} + P + Q_{rad}} \qquad (3)$$

Since $Q_{rad}$ cannot be directly measured, in real measurement $Q_{in}$ is composed of $Q_{out}$ and $P$ which leads to the measurement error of $\eta$. By tuning the current in the circuit, a series of $Q_{in}$, $P$ can be measured at the same time. Therefore, both maximum $\eta$ and $P$ can be found. To minimize the radiation loss, copper foil working as a radiation shield is brazed with copper plate of hot side. Since this radiation shield is at higher temperature than the leg, it will add additional heat flow into the leg, therefore the measured

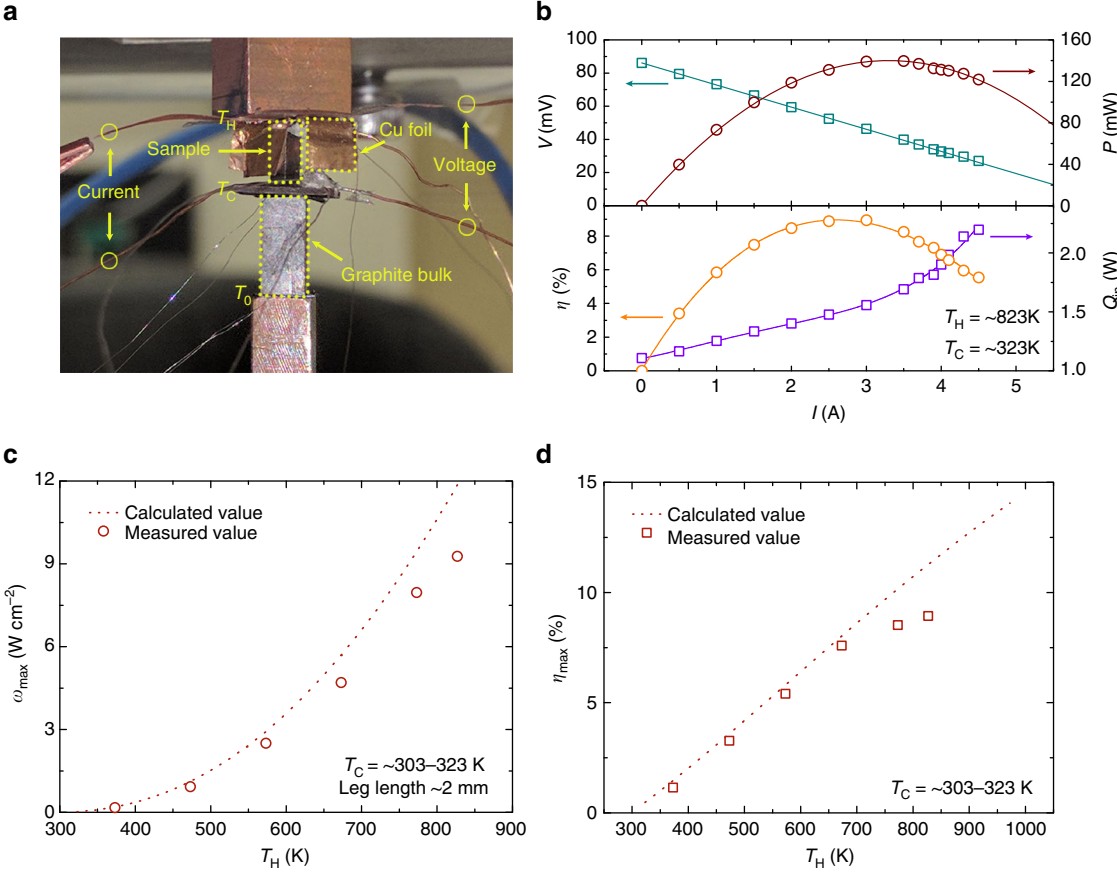

**Fig. 5** Measurement of output power density and thermoelectric conversion efficiency. **a** Experimental setup of a single-leg device. **b** The measured voltage, output power, input power and efficiency of the device with varying current. The cold-side and hot-side temperature of the device are fixed at ~323 and ~823 K, respectively. Hot-side-temperature-dependent output power density (normalized to the length of 2 mm) (**c**) and heat-to-electricity conversion efficiency (**d**). The cold-side temperature is fixed at 298 K for the calculation of output power density and conversion efficiency

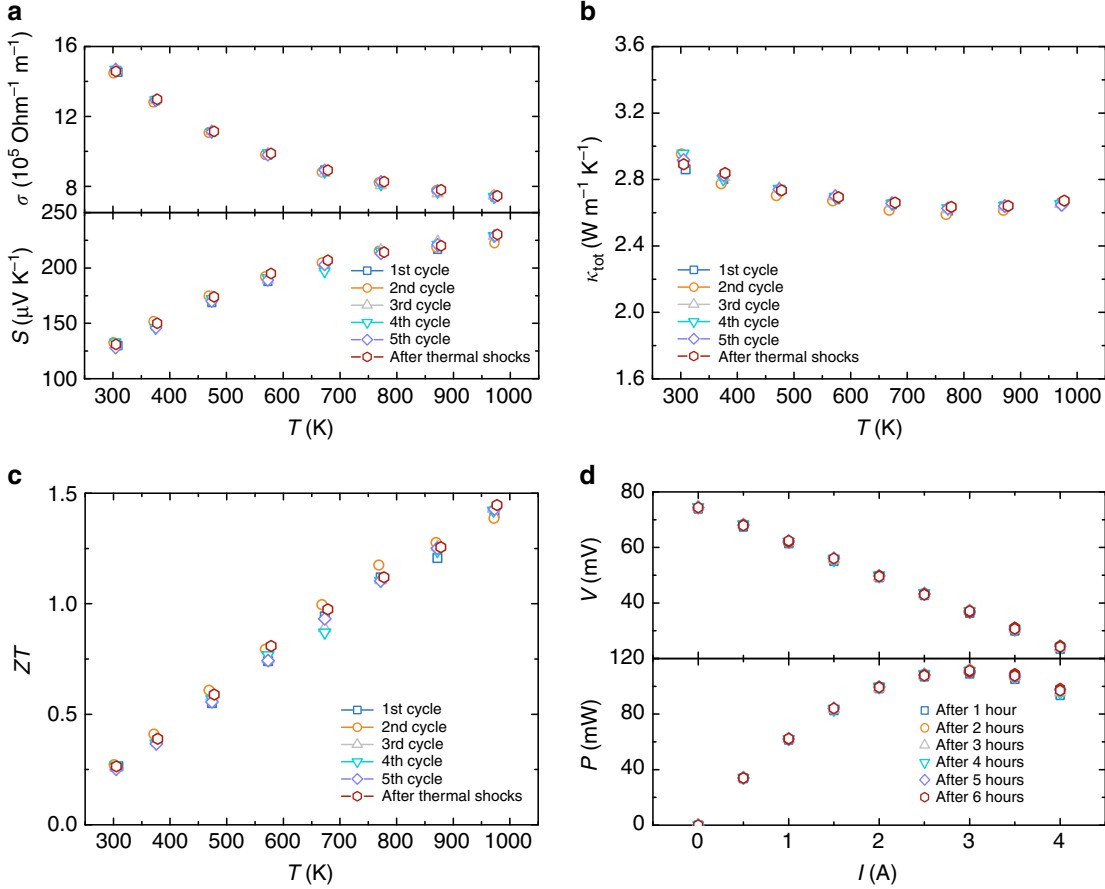

**Fig. 6** Thermal stability test. Repeated measurement of electrical conductivity and Seebeck coefficient (**a**), total thermal conductivity (**b**), ZT (**c**) for the ZrCoBi$_{0.65}$Sb$_{0.15}$Sn$_{0.20}$ samples. Repeated measurement of the current-dependent voltage and output power (**d**) for the ZrCoBi$_{0.65}$Sb$_{0.15}$Sn$_{0.20}$ leg maintained at a cold-side temperature of ~368K and a hot-side temperature ~803K

$Q_{out}$ will actually be higher than without the shield. This should lead to a more conservative value of efficiency.

In this work, all the parameters for the single leg device of ZrCoBi$_{0.65}$Sb$_{0.15}$Sn$_{0.20}$, including temperature difference, electric current ($I$), voltage ($V$), output power ($P$), and input power ($Q_{in}$) can be obtained simultaneously, as shown in Fig. 5b. Due to the limited heating power of the system, the hot-side temperature ($T_H$) can only be raised to ~823 K. The measured hot-side-temperature-dependent maximum output power density ($\omega_{max}$) and maximum efficiency ($\eta_{max}$) are shown in Fig. 5c, d, respectively. The peak output power density is ~9.3 W cm$^{-2}$ and the efficiency is ~9% at the hot-side temperature of ~823 K (Fig. 5d). It is noted that the obtained results are slightly lower than that of the theoretical calculations[41], especially at high temperature. This should be mainly ascribed to the radiation heat, the rise of cold-side temperature, and parasitic electrical and heat loss. By minimizing these adverse effects, it is expected that the measured maximum efficiency and output power density will be closer to the calculated values ~12% and ~11 W cm$^{-2}$ at hot-side temperature of ~823 K. Even higher values of maximum output power density and efficiency can be possibly achieved by increasing the hot-side temperature, as shown in Supplementary Fig. 9.

**Good thermal stability**. Since the potential applications of the half-Heusler materials are usually associated with a high temperature difference and repeat thermal shocks, it is necessary to further verify the thermal stability of the materials. As shown in Fig. 6, the thermal stability for ZrCoBi$_{0.65}$Sb$_{0.15}$Sn$_{0.20}$ has been tested. Repeat thermoelectric measurements between 300 and 973 K for this sample has been conducted and thermoelectric properties remain similar for each cycle (Fig. 6a–c). Afterwards, the sample was then sealed in an evacuated quartz ampoule and directly heated in the furnace with the temperature of 973 K for 10 min and then rapidly cooled to room temperature by air quenching. Such a thermal shock treatment has been repeated for ten times. The thermoelectric properties were then measured again and the results still remain similar. In addition, to evaluate the effect of large temperature gradient on the stability of thermoelectric performance for ZrCoBi-based half-Heuslers, the ZrCoBi$_{0.65}$Sb$_{0.15}$Sn$_{0.20}$ leg has been maintained at a cold-side temperature of ~368 K and a hot-side temperature of ~803 K (corresponding to a large temperature gradient of ~100 K mm$^{-1}$) for 6 h. The current-dependent voltage and output power have been measured for each hour and the results are quite comparable (Fig. 6d). Furthermore, a thermogravimetric analysis was conducted for ZrCoBi$_{0.65}$Sb$_{0.15}$Sn$_{0.20}$ and no decomposition of the sample was observed up to 1273 K in the Ar atmosphere (Supplementary Fig. 10). Therefore, all the results indicate the good thermal stability of the prepared ZrCoBi-based half-Heuslers.

## Discussion

In this work, p-type ZrCoBi-based half-Heuslers with high thermoelectric performance are reported. A record-high ZT ~1.42 at 973 K is achieved in ZrCoBi$_{0.65}$Sb$_{0.15}$Sn$_{0.20}$ that outperforms all the previously report p-type half-Heuslers at the same

temperature. Importantly, the average ZT of ~0.81 (calculated by integration method between 300 and 973 K) for ZrCoBi$_{0.65}$Sb$_{0.15}$Sn$_{0.20}$ is also the highest value among all the p-type half-Heusler compounds. In addition, maximum output power density and conversion efficiency of ~9.3 W cm$^{-2}$ and ~9% were realized with cold-side and hot-side temperature being 323 and 823 K, respectively. In addition, a good thermal stability for the ZrCoBi-based half-Heusler has also been confirmed. Our results demonstrate that ZrCoBi-based half-Heusler compounds are promising high-temperature thermoelectric materials. More importantly, our work indicates that the Bi-based half-Heuslers, which their thermoelectric properties have rarely been investigated previously, have a great potential for realizing high thermoelectric performance.

## Methods

**Synthesis**. The ZrCoBi samples were prepared by ball-milling and hot-pressing method. Pure elements (Zr granules, 99.2%; Co powders, 99.8%; Bi ingots, 99.999%; Sb ingots, 99.999%; and Sn powders, 99.8%; Alfa Aesar) according to the composition of ZrCoBi$_{1-x}$Sn$_x$ (x = 0, 0.05, 0.10, 0.15, 0.20, and 0.25), and ZrCoBi$_{0.80-y}$Sb$_y$Sn$_{0.20}$ (y = 0.05, 0.10, 0.15, 0.20) were loaded in a stainless-steel jar under an argon atmosphere in the glove box. The ball milling process was conducted on SPEX 8000 M Mixer/Mill for 20 h. The ball-milled powders were compacted to disk by a direct current induced hot press at about 1173 K for 5 min and under the pressure of ~50 MPa.

**Transport properties measurement**. The Seebeck coefficient and electrical conductivity were obtained simultaneously by a commercial (ZEM-3, ULVAC) system under a helium atmosphere. The thermal conductivity $\kappa = DC_p\rho$ was calculated from the thermal diffusivity $D$ (Supplementary Fig. 11a and 11b), specific heat $C_p$ (Supplementary Fig. 11c), and mass density $\rho$, which were measured by laser flash (LFA457, Netzsch), a differential scanning calorimeter (DSC 404 C; Netzsch), and an Archimedes' kit, respectively. Hall carrier concentrations $n_H$ were measured on a commercial system (PPMS, Quantum Design), with a magnetic field of ±3 T and an electrical current of 8 mA.

**Microstructural characterization**. Phase identification were carried out by X-ray diffraction (XRD) on a PANalytical multipurpose diffractometer with an X'Celerator detector (PANalytical X'Pert Pro). The morphology and microstructures were characterized by a field emission scanning electron microscope (FESEM, LEO 1525) and a high-resolution transmission electron microscope (HRTEM, JEOL 2010F) as shown in Supplementary Fig. 12. SEM and EDS were performed by energy-dispersive X-ray spectroscopy (JEOL JSM-6330F) as shown in Supplementary Figs. 13-15. Thermogravimetric analysis was carried out by the simultaneous thermal analyzer (NETZSCH STA 449 F3 Jupiter).

**Sound velocity measurement**. Sound velocity measurements were carried out by a RITEC Advanced Ultrasonic Measurement System RAM-5000. The system realizes pulse-echo method of time propagation measurements with an accuracy of about 10$^{-3}$ μs. To generate longitudinal (L) and shear (S) ultrasonic bulk waves, Olympus transducers V129-RM (10 MHz) and V157-RM (5 MHz) were used. Propylene glycol and SWC (both from Olympus) were used as couplant materials for L and S modes, respectively. Thickness measurements were carried out using Mitutoyo ID-HO530 device. All data were obtained at 300 K.

**Theoretical calculation**. The electronic structures were obtained using the linearized augmented plane-wave (LAPW) method as implemented in the WIEN2K code[42]. The experimental lattice constants were fixed for TiCoSb, ZrCoSb, and ZrCoBi and the internal atomic positions were relaxed within the Perdew, Burke, and Ernzerhof (PBE) functional[43] by total energy minimization. Then the modified Becke–Johnson (mBJ) potential[44] was used for the band structure and isosurface calculations. We used LAPW sphere radii of 2.4 Bohr for Ti, Co and Sb, and 2.5 Bohr for Zr and Bi. A basis set cut-off parameter $R_{min}K_{max} = 9$ was used. We used well converged k-point grids for the relaxation and self-consistent calculations, and denser k-meshes in the isosurface calculations. Spin-orbit coupling (SOC) is included in all the calculations except for the structural relaxations.

The lattice thermal conductivity of ZrCoBi$_{1-x}$Sn$_x$ was calculated within the virtual crystal approximation, where we averaged the harmonic and anharmonic inter-atomic force constants (IFCs) of ZrCoBi and ZrCoSn according to the doping level x. The three-phonon scattering of ZrCoBi$_{1-x}$Sn$_x$ was then calculated through the harmonic and anharmonic IFCs[45]. The scattering of the phonons by the Bi/Sn doping effect was included with the mass-variation approximation[46]. The Peierls–Boltzmann equation was then solved iteratively to compute the lattice thermal conductivity at different x and temperatures. All first-principles calculations were carried out in the QUANTUM ESPRESSO package[47] with the

Perdew–Burke–Ernzerhof exchange-correlation functional[43]. We first optimized the lattice constants of ZrCoBi and ZrCoSn in the MgAgAs structure. The harmonic IFCs were then calculated within the density functional perturbation theory[48] as implemented in QUANTUM ESPRESSO on a 6 × 6 × 6 q mesh. The anharmonic IFCs were computed with the finite difference method on a 3 × 3 × 3 supercell. The MFP for a particular phonon mode $\lambda = (q, v)$ is defined as $l_\lambda = |v_\lambda|\tau_{\lambda\alpha}$, where $\tau_{\lambda\alpha}$ is the phonon lifetime at reciprocal vector $q$ and branch index $v$ and $\alpha$ is the Cartesian direction.

**Data availability**. The data that support the findings of this study are available from the corresponding author upon reasonable request.

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

## Acknowledgements
The work is supported by the Solid-State Solar-Thermal Energy Conversion Center (S³TEC), an Energy Frontier Research Center funded by the U.S. Department of Energy, Office of Science, Office of Basic Energy Sciences under Award Number: DE-SC0001299. D.B. and C.L. also acknowledge support from the National Science Foundation under Grant No. 1402949. Y.W. also acknowledges National Natural Science Foundation of China (Grant No.11474329).

## Author contributions
H.Z. and Z.R. designed the research, H.Z., R.H., and J.M. performed the experiments, Q. Z. measured the efficiency and output power, C.L., D.B., J.S., and D.S. conducted the first-principles calculations, Y.W. conducted the TEM characterization, Z.T. helps with the XRD measurement, A.S. measured the sound velocity, W.R., Z.L., Z.W., G.C., and K.N. contributed analytical tools, H.Z., J.M., and Z.R. wrote the manuscript and everyone provided the comments.

## Additional information

**Competing interests:** The authors declare no competing interests.

