## [Peer Review File · Nature Communications]

Reviewers' comments:

Reviewer #1 (Remarks to the Author):

The manuscript by Ren and Chen, et al, reported the discovery of ZrCoBi-based half-Heuslers with high thermoelectric conversion efficiency. The only thing missing, which is extremely critical, as everyone should expect from two well-known groups in the field of thermoelectric, is to characterize the long-term stability of their new materials, if the authors want to claim the "real" application of it. The authors only cited a publication by other group, but didn't show any data for the new material's thermal decomposition in air or in vacuum. Also, it is unclear how stable this material is when facing a long-term large temperature gradient and a large thermal shock.

I suggest to accept the manuscript only after these missing critical data are presented.

Reviewer #2 (Remarks to the Author):

This manuscript reports the discovery of ZrCoBi-based half-Heuslers with high band degeneracy and low mean sound velocity. The electronic band structure of three half-Heusler compounds were studied, namely, ZrCoBi, ZrCoSb, TiCoSb. Of the three, ZrCoBi and ZrCoSb show high band degeneracy ($N_v = 10$ and $N_v = 8$, respectively), which suggest that high power factors may be achievable. Samples of ZrCoBi-based materials were synthesized via ball milling and hot-pressing, and show single-phase XRD patterns with an average grain size of about 200 nm. The specific series of materials studied are a solid solution of ZrCoBi and ZrCoSn, where increasing Sn concentration increased electrical conductivity due to increased carrier concentration. Pure ZrCoBi shows n-type behavior, but increasing Sn up to 20 at % shows successful and efficient doping to p-type behavior. Additionally, bipolar conduction is suppressed by incorporation of Sn.

When compared to Sn-doped TiCoSb and ZrCoSb, the authors show a correlation between high power factor and high band degeneracy. However, the authors did not state whether the carrier concentrations for all three Sn-doped samples were similar in magnitude. Since Seebeck is highly dependent on carrier concentration, it is suggested that the carrier concentration be stated to reduce ambiguity in the correlation between power factor and band degeneracy.

Thermal conductivity for all samples decrease as a function of Sn concentration, where it is lowest for ZrCoBi_{0.80}Sn_{0.20}. The authors attribute this to point-defect scattering at Sn sites. This results in a peak ZT of 1.3 at 973 for ZrCoBi_{0.80}Sn_{0.20}. ZT can be further improved by reducing the thermal

conductivity with Sb alloying, resulting in a peak ZT of 1.42 at 973 K for $\text{ZrCoBi}_{0.65}\text{Sb}_{0.15}\text{Sn}_{0.20}$. These samples show excellent reproducibility.

Lastly, thermoelectric efficiency was measured which showed a conversion efficiency of 9 % at 823 K for $\text{ZrCoBi}_{0.65}\text{Sb}_{0.15}\text{Sn}_{0.20}$. The authors report that this is the highest efficiency for half-Heuslers, where increased efficiency (up to 11 %) can be achieved by reducing parasitic heat and electrical losses.

Overall, the manuscript presents an excellent research on half heusler $\text{ZrCoBi}_{1-x}\text{Sn}_x$ with high performance for thermoelectric application. However, there are still many aspects that need to be clarified to support the theory adopted to explain all transport properties. I have listed some questions for example as following:

1. These materials were synthesized by ball milling in stainless-steel Jar, I wonder how severe is the Fe contamination? Fe contamination is a common problem in ball milling synthesis. What influences could Fe impurity bring on electrical and thermal conductivity? No micro information about dispersion of chemical elements is given. When doping at Bi site with Sn up to 20 at%, how phase constituents change? Is there phase separation since the lattice parameter display non-linear change with respect to Sn concentration.
2. Ball milled samples show refined grains, how to differentiate effect of decreased grain size, Sn doping at Bi sites and the so called low sound velocity? Did Sn doping significantly influence microstructure of materials? No information gives such comparison.
3. Indeed, the measured efficiency is high. What's the error for this measurement on home-built system? As for highest efficiency compared to other published result, is the comparison made by comparing result from similar measurement, for instance all based on single leg? If not, I would not accept such comparison because things get complicated once more legs adopted, for example thermal & electrical contact.
4. It is good to see consistent performance among different batch of samples. How is the stability of performance with respect to heating circle? Will refined grains suffer coarsening during service or even repeated measurements?
5. Reporting carrier concentrations for Figure 3D for a more complete picture of the contributions to power factor.

6. Did the authors perform any EDS mapping of the materials to confirm the phase purity? Ball milling for 20 hours in steel jars may result in significant Fe contamination.

Overall, I would recommend publication if such details as listed above are well clarified, making the whole story sound more reasonable.

Reviewer #3 (Remarks to the Author):

The paper by Hangtian Zhu et al. on the discovery and characterization of a novel high ZT TE material ZrCoBi is an original work of high quality. It is well organized and well written. Arguments are sound - Thus the paper should be published essentially as it stands - only the statements on the status of the art on high ZT half Heusler alloys are misleading the reader and so is Figure 1. A quick check in science direct on high ZT half Heusler alloys prompts a set of recently published results (particularly with alloys based on TiNiSn) that have ZT reaching $ZT=1.3$ and even $ZT=1.5$ already below 900 K. These curves should in any case be added to Fig 1 prior to publication (adjusting the corresponding statements in the text):

1) High thermoelectric figure of merit by resonant dopant in Half-Heusler alloys; By Chen, Long; Liu, Yamei; He, Jian; Tritt, Terry M.; Joseph, Poon S. From arXiv.org, e-Print Archive, Condensed Matter (2017), 1-11. | Language: English, Database: CAPLUS

2) (V,Nb)-doped half Heusler alloys based on {Ti,Zr,Hf}NiSn with high ZT By Rogl, G.; Sauerschnig, P.; Rykavets, Z.; Romaka, V. V.; Heinrich, P.; Hinterleitner, B.; Grytsiv, A.; Bauer, E.; Rogl, P. From Acta Materialia (2017), 131, 336-348. | Language: English, Database: CAPLUS

Reviewer #1

Comment: The manuscript by Ren and Chen, et al, reported the discovery of ZrCoBi-based half-Heuslers with high thermoelectric conversion efficiency. The only thing missing, which is extremely critical, as everyone should expect from two well-known groups in the field of thermoelectric, is to characterize the long-term stability of their new materials, if the authors want to claim the "real" application of it. The authors only cited a publication by other group, but didn't show any data for the new material's thermal decomposition in air or in vacuum. Also, it is unclear how stable this material is when facing a long-term large temperature gradient and a large thermal shock. I suggest to accept the manuscript only after these missing critical data are presented.

Response: Thank you very much for the comments. Systematic studies on the thermal stability of ZrCoBi-based materials have been conducted. As shown in Fig. 1, the thermoelectric transport properties have been repeatedly measured. During these measurements, all the thermoelectric properties remained almost the same. In addition, the samples were sealed in evacuated quartz ampoule for the thermal shock treatments. The quartz ampoule was rapidly heated in the furnace with the temperature of 973 K and held for ten minutes and then rapidly cooled to room temperature by air quenching. Such a thermal shock treatment has been carried out for ten cycles. The measured transport properties after the thermal shock treatments do not show any noticeable changes.

Figure 1. Electrical conductivity (a), Seebeck coefficient (b), total thermal conductivity (c), and ZT value (d) of $\text{ZrCoBi}_{0.65}\text{Sb}_{0.15}\text{Sn}_{0.20}$ for repeated measurements and after the thermal shock treatments.

Additionally, the thermogravimetric analysis of $\text{ZrCoBi}_{0.65}\text{Sb}_{0.15}\text{Sn}_{0.20}$ was conducted in Ar atmosphere and no decomposition of the sample was observed up to 1273 K (as shown in Fig. 2).

Figure 2. Thermogravimetric analysis of $\text{ZrCoBi}_{0.65}\text{Sb}_{0.15}\text{Sn}_{0.20}$ in Ar atmosphere.

In order to further validate the long-term stability of the thermoelectric performance under large temperature gradient, a temperature gradient of ~ 100 K/mm has been maintained on the single leg with a homemade system for six hours. The current-dependent voltage and output power have been measured for each hour, as shown in Fig. 3. No noticeable changes were observed in the long-term measurement. All the results here are included in the revised Supporting Information.

Figure 3. The long-term stability of $\text{ZrCoBi}_{0.65}\text{Sb}_{0.15}\text{Sn}_{0.20}$ single-leg under large temperature gradient.

Reviewer #2

This manuscript reports the discovery of ZrCoBi-based half-Heuslers with high band degeneracy and low mean sound velocity. The electronic band structure of three half-Heusler compounds were studied, namely, ZrCoBi, ZrCoSb, TiCoSb. Of the three, ZrCoBi and ZrCoSb show high band degeneracy ($N_v = 10$ and $N_v = 8$, respectively), which suggest that high power factors may be achievable. Samples of ZrCoBi-based materials were synthesized via ball milling and hot-pressing, and show single-phase XRD patterns with an average grain size of about 200 nm. The specific series of materials studied are a solid solution of ZrCoBi and ZrCoSn, where increasing Sn concentration increased electrical conductivity due to increased carrier concentration. Pure ZrCoBi shows n-type behavior, but increasing Sn up to 20 at % shows successful and efficient doping to p-type behavior. Additionally, bipolar conduction is suppressed by incorporation of Sn.

Thermal conductivity for all samples decrease as a function of Sn concentration, where it is lowest for ZrCoBi_{0.80}Sn_{0.20}. The authors attribute this to point-defect scattering at Sn sites. This results in a peak ZT of 1.3 at 973 for ZrCoBi_{0.80}Sn_{0.20}. ZT can be further improved by reducing the thermal conductivity with Sb alloying, resulting in a peak ZT of 1.42 at 973 K for ZrCoBi_{0.65}Sb_{0.15}Sn_{0.20}. These samples show excellent reproducibility.

Lastly, thermoelectric efficiency was measured which showed a conversion efficiency of 9 % at 823 K for ZrCoBi_{0.65}Sb_{0.15}Sn_{0.20}. The authors report that this is the highest efficiency for half-Heuslers, where increased efficiency (up to 11 %) can be achieved by reducing parasitic heat and electrical losses.

Overall, the manuscript presents an excellent research on half heusler ZrCoBi_{1-x}Sn_x with high performance for thermoelectric application. However, there are still many aspects that need

to be clarified to support the theory adopted to explain all transport properties. I have listed some questions for example as following:

Comment: When compared to Sn-doped TiCoSb and ZrCoSb, the authors show a correlation between high power factor and high band degeneracy. However, the authors did not state whether the carrier concentrations for all three Sn-doped samples were similar in magnitude. Since Seebeck is highly dependent on carrier concentration, it is suggested that the carrier concentration be stated to reduce ambiguity in the correlation between power factor and band degeneracy.

Response: Thank you very much for the comments. In the revised manuscript, all the $\text{TiCoSb}_{0.85}\text{Sn}_{0.15}$, $\text{ZrCoSb}_{0.85}\text{Sn}_{0.15}$ and $\text{ZrCoBi}_{0.85}\text{Sn}_{0.15}$ samples are synthesized by the ball milling method, and electrical transport properties have been characterized. The Hall carrier concentration for 15 % Sn-doped TiCoSb, ZrCoSb, and ZrCoBi are 1.62×10^{21} , 1.47×10^{21} , and $2.20 \times 10^{21} \text{ cm}^{-3}$, respectively. With the similar Hall carrier concentration, the relation between power factor and band degeneracy in the half-Heulser system can be clearly identified.

Comment: These materials were synthesized by ball milling in stainless-steel Jar, I wonder how severe is the Fe contamination? Fe contamination is a common problem in ball milling synthesis. What influences could Fe impurity bring on electrical and thermal conductivity? No micro information about dispersion of chemical elements is given. When doping at Bi site with Sn up to 20 at%, how phase constituents change? Is there phase separation since the lattice parameter display non-linear change with respect to Sn concentration.

Response: Thank you very much for your comments. EDS mapping has been carried out to investigate the Fe contamination in different samples. As shown in Fig. 4-6, Fe peaks in EDS spectrum are very weak, especially for the sample with Sb alloying where the Fe peaks almost cannot be distinguished even in the enlarged image (Fig. 6h). Therefore, the Fe contamination is not a major issue for the thermoelectric performance of the prepared ZrCoBi-based materials.

Figure 4. (a) SEM image of the hot-pressed ZrCoBi_{0.95}Sn_{0.05} sample. The corresponding EDS elemental mapping of Zr (b), Co (c), Bi (d), Sn (e), and EDS spectrum (f).

Figure 5. (a) SEM image of the hot-pressed $\text{ZrCoBi}_{0.80}\text{Sn}_{0.20}$ sample. The corresponding EDS elemental mapping of Zr (b), Co (c), Bi (d), Sn (e), and EDS spectrum (f).

Figure 6. (a) SEM image of the hot-pressed $\text{ZrCoBi}_{0.65}\text{Sb}_{0.15}\text{Sn}_{0.20}$ sample. The corresponding EDS elemental mapping of Zr (b), Co (c), Bi (d), Sn (e), and Sb (f). EDS spectrum (g) and enlarged EDS spectrum focused on Fe peak (h). The analysis is taken from the sample after repeated measurement and ten times thermal shock cycles from room temperature to 973 K (Figure S13).

As shown in the EDS mapping (Fig. 4-6), all the elements are uniformly distributed. No phase separation can be observed in the samples. Therefore, phase separation should not exist in the samples. Lattice parameters of the alloy can be roughly estimated by Vegard's law which describe a linear relation between the lattice parameter and the concentration. However, this law is only an empirical rule. In fact, deviation from linear behavior is often observed in other material systems (A. R. Denton and N. W. Ashcroft, Physical Review A, 1991, 43, 3161-3164; V. A. Lubarda, Mech. Mater., 2003, 35, 53-68). The exact reason for the deviation from the Vegard's law is complicated and will be studied in the future.

Comment: Ball milled samples show refined grains, how to differentiate effect of decreased grain size, Sn doping at Bi sites and the so called low sound velocity? Did Sn doping significantly influence microstructure of materials? No information gives such comparison.

Response: Thanks you very much for your comments. As observed in the SEM images (Fig. 7), Sn doping and Sb alloying can greatly reduce the average grain size. The average grain size is ~2 μm for undoped ZrCoBi, ~400 nm for ZrCoBi_{0.95}Sn_{0.05}, and ~200 nm for ZrCoBi_{0.65}Sb_{0.15}Sn_{0.20}.

Figure 7. SEM images of hot-pressed $\text{ZrCoBi}_{1-x-y}\text{Sb}_y\text{Sn}_x$ with different doping and alloying concentration. Undoped ZrCoBi (a), $\text{ZrCoBi}_{0.95}\text{Sn}_{0.05}$ (b), $\text{ZrCoBi}_{0.90}\text{Sn}_{0.10}$ (c), $\text{ZrCoBi}_{0.85}\text{Sn}_{0.15}$ (d), $\text{ZrCoBi}_{0.80}\text{Sn}_{0.20}$ (e) and $\text{ZrCoBi}_{0.65}\text{Sb}_{0.15}\text{Sn}_{0.20}$ (f).

Therefore, the alloying scattering, grain boundary scattering, and intrinsic low sound velocity will jointly contribute to the low lattice thermal conductivity of $\text{ZrCoBi}_{0.65}\text{Sb}_{0.15}\text{Sn}_{0.20}$. In fact, the low sound velocity leads to the low lattice conductivity of ZrCoBi , as shown in the comparison of the thermal conductivity between undoped ZrCoBi with TiCoSb , NbFeSb , and ZrCoSb (Figure 4b in the manuscript). In addition, the alloying scattering and grain boundary scattering will both contribute to the suppression of lattice thermal conductivity in Sn-doped and Sb-alloyed samples. The contributions from the different phonon scattering mechanisms can be estimated from the calculated phonon mean-free-path dependent lattice thermal conductivity (Figure 4d in the manuscript). The calculated accumulative lattice thermal conductivity (with the phonon mean-free-path smaller than 1000 nm) is $\sim 14.7 \text{ W m}^{-1} \text{ K}^{-1}$ for the undoped ZrCoBi and only $\sim 4.9 \text{ W m}^{-1} \text{ K}^{-1}$ for $\text{ZrCoBi}_{0.80}\text{Sn}_{0.20}$. This indicates that the alloying scattering play a dominant role in reducing the lattice thermal conductivity. In addition, the contributions of phonon with mean-free-path from 200 nm to 1 μm are quite negligible for undoped ZrCoBi and $\text{ZrCoBi}_{0.80}\text{Sn}_{0.20}$, suggesting the grain boundary scattering is not as effective as the alloying scattering for reducing the lattice thermal conductivity.

Comment: Indeed, the measured efficiency is high. What's the error for this measurement on home-built system? As for highest efficiency compared to other published result, is the comparison made by comparing result from similar measurement, for instance all based on single leg? If not, I would not accept such comparison because things get complicated once more legs adopted, for example thermal & electrical contact.

Response: Thank you very much for your comments. The total error for the efficiency measurement is ~13% at lower temperature and increases to ~20% at higher temperature (above ~700 K). The error for the measurement of electrical transport properties (arising from deviation of leg temperature distribution from the ideal condition, the Seebeck effect of the copper wire, measurement of temperature, and the soldering) is smaller than ~10%. The error on the thermal conductivity is ~3% at lower temperature which can be attributed to the error of the temperature measurement on the bulk polycrystalline graphite, and increases to ~10% due to the increased radiation and the parasitic heat loss at higher temperature (above ~700 K). In fact, we used the same system for measurements of other known materials. The obtained results are similar with the reported data, which all based on the single leg. In addition, the comparison of the measured efficiency has been removed in the revised manuscript.

Comment: It is good to see consistent performance among different batch of samples. How is the stability of performance with respect to heating cycle? Will refined grains suffer coarsening during service or even repeated measurements?

Response: Systematic studies on the thermal stability of the ZrCoBi-based materials have been conducted. As shown in Fig. 1-3, the thermoelectric properties remain similar after the repeated cycles and thermal shock treatments, indicating good thermal stability. Fig. 8 shows the SEM images of the as-prepared $\text{ZrCoBi}_{0.65}\text{Sb}_{0.15}\text{Sn}_{0.20}$ and after the repeated measurement. Clearly, the average grain size remains similar before and after the repeated measurement,

Figure 8. SEM images of the as-prepared $\text{ZrCoBi}_{0.65}\text{Sb}_{0.15}\text{Sn}_{0.20}$ (a) and after the repeated measurement (b).

Comment: Reporting carrier concentrations for Figure 3d for a more complete picture of the contributions to power factor.

Response: Thank you very much for the suggestion. The Hall carrier concentrations for the samples are included in the revised manuscript.

Comment: Did the authors perform any EDS mapping of the materials to confirm the phase purity? Ball milling for 20 hours in steel jars may result in significant Fe contamination.

Response: Thank you very much for the comments. EDS mappings have been conducted for $\text{ZrCoBi}_{0.95}\text{Sn}_{0.05}$, $\text{ZrCoBi}_{0.80}\text{Sn}_{0.20}$, and $\text{ZrCoBi}_{0.65}\text{Sb}_{0.15}\text{Sn}_{0.20}$ (as shown in Figs. 4-6). Clearly, the Fe peaks in the EDS spectra are very small.

Reviewer #3

The paper by Hangtian Zhu et al. on the discovery and characterization of a novel high ZT TE material ZrCoBi is an original work of high quality. It is well organized and well written. Arguments are sound - Thus the paper should be published essentially as it stands - only the statements on the status of the art on high ZT half Heusler alloys are misleading the reader and so is Figure 1. A quick check in science direct on high ZT half Heusler alloys prompts a set of recently published results (particularly with alloys based on TiNiSn) that have ZT reaching $ZT=1.3$ and even $ZT=1.5$ already below 900 K. These curves should in any case be added to Fig 1 prior to publication (adjusting the corresponding statements in the text):

- 1) High thermoelectric figure of merit by resonant dopant in Half-Heusler alloys; By Chen, Long; Liu, Yamei; He, Jian; Tritt, Terry M.; Joseph, Poon S. From arXiv.org, e-Print Archive, Condensed Matter (2017), 1-11. | Language: English, Database: CAPLUS
- 2) (V,Nb)-doped half Heusler alloys based on TiNiSn with high ZT By Rogl, G.; Sauerschnig, P.; Rykavets, Z.; Romaka, V. V.; Heinrich, P.; Hinterleitner, B.; Grytsiv, A.; Bauer, E.; Rogl, P. From Acta Materialia (2017), 131, 336-348. | Language: English, Database: CAPLUS

Response: Thank you very much for your positive comments. These references are important on half-Heusler materials, we have now cited all the references in the revised manuscript. Since we mainly focused on the *p*-type ZrCoBi in this work, the thermoelectric performance was only compared between $ZrCoBi_{0.65}Sb_{0.15}Sn_{0.2}$ and other state-of-the-art *p*-type half-Heuslers in the figure.

REVIEWERS' COMMENTS:

Reviewer #1 (Remarks to the Author):

Nice studies on stability, no more questions.

Reviewer #2 (Remarks to the Author):

The Authors took adequate steps to properly address all observations from the previous review and I am happy with the additional explanation and experiments provided to support the conclusion of this study.

Reviewer #3 (Remarks to the Author):

The revised version addressed all points remarked in the review. The paper can be published as it stands now.